# Efficiency of Prolonged Prone Positioning for Mechanically Ventilated Patients Infected with COVID-19

**DOI:** 10.3390/jcm10132969

**Published:** 2021-07-01

**Authors:** Elizabeth M. Parker, Edward A. Bittner, Lorenzo Berra, Richard M. Pino

**Affiliations:** 1Surgical Intensive Care Unit, Massachusetts General Hospital, Boston, MA 02114, USA; emparker91@gmail.com; 2Department of Anesthesia, Critical Care and Pain Medicine, Massachusetts General Hospital, Boston, MA 02114, USA; ebittner@mgh.harvard.edu (E.A.B.); lberra@mgh.harvard.edu (L.B.); 3Louisiana State University Health Sciences Center, Department of Anesthesiology, New Orleans, LA 70112, USA

**Keywords:** acute respiratory distress syndrome, ARDS, COVID-19, respiratory failure, prone positioning, PaO_2_/FIO_2_ ratio, hypoxemia

## Abstract

Hypoxemia of the acute respiratory distress syndrome can be reduced by turning patients prone. Prone positioning (PP) is labor intensive, risks unplanned tracheal extubation, and can result in facial tissue injury. We retrospectively examined prolonged, repeated, and early versus later PP for 20 patients with COVID-19 respiratory failure. Blood gases and ventilator settings were collected before PP, at 1, 7, 12, 24, 32, and 39 h after PP, and 7 h after completion of PP. Analysis of variance was used for comparisons with baseline values at supine positions before turning prone. PP for >39 h maintained PaO_2_/FiO_2_ (P/F) ratios when turned supine; the P/F decrease at 7 h was not significant from the initial values when turned supine. Patients turned prone a second time, when again turned supine at 7 h, had significant decreased P/F. When PP started for an initial P/F ≤ 150 versus P/F > 150, the P/F increased throughout the PP and upon return to supine. Our results show that a single turn prone for >39 h is efficacious and saves the burden of multiple prone turns, and there is no significant advantage to initiating PP when P/F > 150 compared to P/F ≤ 150.

## 1. Introduction

In the acute respiratory distress syndrome (ARDS) prone positioning (PP) is commonly used to increase oxygenation with the overall goal to minimize ventilator induced lung injury [1,2,3,4]. The prone position enables a better distribution of transpulmonary pressures, relieves pressure on the lung that is posterior to the heart, and improves lymphatic drainage [1]. It became clear during the initial stages of the COVID-19 pandemic that the ARDS seen in infected patients, termed CARDS, was often more challenging to manage than that seen in other non-COVID-19 conditions [5]. Intensivists found that the initial protocols for the treatment of COVID-19 were arbitrary and routinely changed as new data became available [6,7]. However, the use of PP remained a valuable option for the treatment of ARDS from any etiology [3].

Prone positioning is labor intensive and involves multiple caregivers to turn patients, some of whom may have high body mass indexes. It has inherently increased risks of inadvertent tracheal extubation, endotracheal tube obstruction, facial tissue injury even with optimal precautions, and tracheal stenosis [3,8,9,10]. Initial studies of PP identified an improvement in oxygenation but failed to identify a mortality benefit [8,11,12]. Subsequent reports indicated that PP decreased mortality for patients with severe ARDS, although improved oxygenation may not correlate with mortality [2,3,13,14]. Improvement in oxygenation after the initial PP is a significant predictor of ICU survival and the duration of PP may be an important determinant of its efficacy [14,15].

The treatment of patients with CARDS by experienced intensivists and nurses in a single ICU setting enabled us to examine the impact of duration and factors associated with prone positioning and turning patients back to the supine position, with the goal to determine the most efficient positioning regimen to increase oxygenation and decrease the potential for complications.

## 2. Materials and Methods

With the approval of the Institutional Review Board, we retrospectively examined the results of repeated prone positioning for 20 patients with respiratory failure secondary to COVID-19 infections from March to May 2020 in the Surgical Intensive Care Unit at the Massachusetts General Hospital at the height of the COVID-19 pandemic surge. A total of 25 patients with COVID-19 pneumonia were admitted during this period. Five were excluded from the study because they were stable enough not to require prone positioning. Pharmacologic treatment of COVID-19 was standardized and included hydroxychloroquine, azithromycin, and atorvastatin as per institutional guidelines in use at the time.

This ICU is staffed 24/7 by critical care anesthesiologists, surgeons, and experienced nurses. Attending intensivists were directly involved with the decisions to turn prone and back to supine positions that were based on clinical judgment using P/F ratios after mechanical ventilation parameters (best positive end expiratory pressure (PEEP), driving pressures, neuromuscular blockade and sedation) were optimized. Best PEEP was determined by recruitment maneuvers or stepwise increases in PEEP [16,17] for patients with unstable hemodynamics. Prone positioning was done either by the ICU team or a hospital organized proning team. Patients in the prone position underwent intermittent head turns to prevent skin breakdown. Nasogastric tube feeding was continued in the prone position if tolerated.

Three distinct groups were selected for analyses: prolonged PP, repeated PP, and an early proning (P/F ratio > 150) versus later proning (P/F ratio ≤ 150). Blood gas values and mechanical ventilator settings were studied when they were obtained before PP; at 1, 7, 12, 24, 32, and 39 h after PP initiation; and at 7 h after completion of PP. Since this is a retrospective study, the times were chosen based on the average the closest available data points in the setting of reduced availability to obtain arterial blood gases secondary to short supplies of blood gas syringes during the pandemic.

### Statistical Analysis

Descriptive statistics were performed and reported as mean and standard deviation for continuous variables or as percentages for categorical data as appropriate. Comparisons of the differences between prone and supine positioning at different time points were performed using repeated measures analysis of variance. Bonferroni correction was used to account for multiple comparisons. A two-sided *p*-value of ≤0.05 was used to denote statistical significance. Statistical analyses were performed using Excel^®^ version 1908 (Microsoft^®^ Excel^®^ for Office 365 MSO, Redmond, WA, USA).

## 3. Results

Table 1 summarizes the timeline from symptom onset to the first prone positioning.

### 3.1. Prolonged Prone Positioning

Twelve patients experienced 16 episodes of prolonged PP (>39 h) (Table 2) with ventilation and oxygenation data for the first 39 h presented in Table 3.

The average change in the P/F ratio over the 39 h is illustrated in Figure 1. Compared with the mean P/F ratio prior to PP (130 ± 28.4), significant improvement was found at 1, 7, 12, 24, 32, and 39 h after the PP start. There was no significant improvement between the mean P/F ratio at 12 h and 24 h (*p* = 0.61). The gain in oxygenation over the prolonged PP was maintained when compared to the mean P/F ratio prior to the PP completion, with no significant decreases at 7 h after returning prone (Figure 2). Post hoc analysis showed a significant improvement in the P/F ratio that was evidenced immediately after PP initiation at 1 h, with evidence of ongoing improvement seen between 12–24 h. The P/F values remained constant up to 39 h thereafter. Following completion of prolonged PP, two patients were extubated, six patients received a tracheostomy, and four patients died.

### 3.2. Episodes of Prone Positioning Alternated with Supination

Twelve patients experienced episodes of repeated PP (reproning). Ten patients were male, mean age 57 ± 16.3 years. Table 4 contains ventilation and oxygenation parameters for reproning. The changes in the P/F ratio for reproning are illustrated in Figure 3, Figure 4, Figure 5 and Figure 6. Figure 3 and Figure 4 demonstrate failures to improve after less than 39 h in the prone position but showed increased P/F ratios after reproning for 51 h (Figure 3) and 93 h (Figure 4). Collectively, both the first proning and reproning maintained oxygenation as there was no decrease post completion of PP (Figure 5 and Figure 6). Following completion of PP and turning supine, three patients were extubated, five needed tracheostomies for prolonged ventilatory failure, and four patients died.

### 3.3. Early Versus Late PP

Ten patients underwent “early” PP when the P/F ratio was >150 (six men, mean age 61 ± 19.6 years). Ten patients (seven men, age 57 ± 22.1 years) were placed in PP when the P/F ratio declined to ≤150. The mean change in P/F ratio for positioning prone later versus early is illustrated in Figure 7. There was an increase in the P/F ratio when initiating PP when the P/F ratio ≤150 which was (117 vs. 175, *p* = 0.0004). However, there was not a significant increase in the mean P/F ratio when initiating PP when the P/F ratio >150 (188 vs. 192, *p* > 0.05). After early PP, three patients were extubated, three patients received a tracheostomy, and four died. Two of the patients who died had single proning durations of 23 and 32 h. Following completion of later PP, two patients were extubated, five received a tracheostomy, and three died.

### 3.4. Adverse Events

During the course of this study period there were no inadvertent extubations or soft tissue injuries to the face related to the prone positioning.

## 4. Discussion

Our findings show that a single PP for >39 h is efficacious and saves the labor-intensive burden of multiple supine to prone turns. We have also demonstrated that there is no significant advantage to starting the PP when P/F was >150 compared to P/F ≤ 150. Prior studies of PP for ARDS have focused on the benchmarks of mortality or improvements in oxygenation [2,11,12,15,18,19]. Many studies were published prior to the established definition of ARDS by the Berlin Criteria and the etiologies for hypoxemia were diverse and included pneumonia, sepsis, trauma, and unknown causes [8,11,12,15,18,20,21,22,23]. Exclusions included the use of high-dose vasopressors as commonly seen in many pathophysiologic states concurrent with ARDS [18]. Criteria for turning patients prone often were not well defined and were inconsistent with respect to the time course for PP, the number of times PP was used for each patient, and the total duration of the PP (Appendix A). Ventilation strategies of earlier studies were before the advent of best PEEP [16,17]. Pre-COVID studies recommended PP for at least 12 h/day with the caveat that the PP is associated with tracheal tube obstruction and pressure sores [3,13]. For patients who were first turned prone as long as nine days after the diagnosis of ARDS, the duration of PP for patients was suggestive of an important determinant of effectiveness [15]. Progressive increases in oxygenation were seen after 18 h of prone positioning but the P/F ratio for study entry was <300, which was higher than previously reported [24]. For patients who were in the prone position for greater than 40 h, oxygenation improved between 8 to 16 h without further increases with time [25]. This is in contrast to our findings that showed continued increases in the P/F ratio up to 39 h during a single prone position. For trauma patients with ARDS, the effects on oxygenation with prone positioning were lost during short periods of supine positioning [26]. This gives credence to our findings that prolonged prone positioning is beneficial.

The World Health Organization’s guidance for COVID-19 patients with P/F ratios <150 is prone positioning for 12–16 h per day [27]. A retrospective study of 10 COVID-19 patients suggested that sustained improvement in oxygenation can be only achieved after several cycles of PP with extended times beyond 16 h at the cost of work overload [28]. One prospective study demonstrated that PP improved oxygenation when instituted early, but there were only eight patients requiring mechanical ventilation and the definition of early prone and duration were not defined [29]. There were no significant improvements for awake patients supported with non-invasive positive pressure ventilation who could turn themselves prone, but only half (23 subjects), called “responders,” maintained improved oxygenation.

One limitation of our retrospective study was that the findings were for twenty patients. While several pre-COVID trials included 100 subject or more, our cohort is similar to those of other COVID and non-COVID ARDS investigations (Table 4) [2,8,11,12,14,20,21,23,26,27,28,29,30,31,32]. The strength of our study is that PP was done for patients with a single etiology for ARDS, without exclusions, and with PP done by a dedicated small number of experienced physicians and nurses with similar practice patterns. During the COVID-19 “surge” patients were treated in intensive care units and hospital floors by caregivers who were often not trained in intensive care medicine. Given the differences in treatment, these patients would not have been valid for inclusion in our work. This is in concert with studies of focus groups that identified knowledge, resources, and team culture among the successful determinants for prone positioning [33,34]. In our retrospective study design, we cannot determine if the maintenance of the P/F ratio was due to the positioning itself versus the progress of natural recovery that was allowed secondary to the increase in oxygenation during the prolonged prone position duration.

Our findings demonstrate that a single prone turn for >39 h improves oxygenation. In addition to reducing the work effort by caregivers and exposure to the pathogen, the risk of an unplanned extubation of a hypoxemic patient is minimized. Since the prone turn for P/F ratios ≤150 shows improvement in oxygenation, the decision to change positioning from supine to prone can be delayed until there are additional clinical data to support the turn and resources available for turning support.

There is evidence that a change from supine to prone position increases endotracheal tube cuff pressures, the result of which may cause tracheal stenosis [10]. It is likely that even with judicious monitoring and adjustment of cuff pressures, frequent changes in position in an attempt to increase the P/F ratio may result in damage to the tracheal mucosa. Our results support the limitation of supine to prone turns that should minimize the risk of tracheal injury.

## Figures and Tables

**Figure 1 jcm-10-02969-f001:**
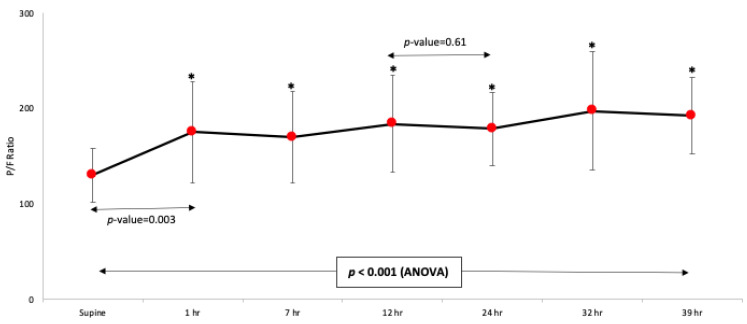
Time course change in P/F ratio during the first 39 h of prolonged prone position. Supine represents positioning before initiating prone positioning. Compared to the mean P/F ratio prior to PP, significant improvement was found at 1, 7, 12, 24, 32, and 39 h after PP.

**Figure 2 jcm-10-02969-f002:**
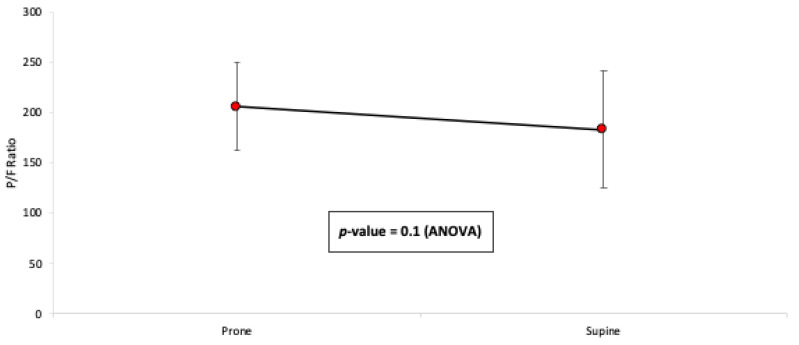
Mean P/F ratio before and after proning position completion. Prone represents time before turning back to the supine position. Supine represents 7 h after completing PP. The gain in oxygenation over the prolonged PP for >39 h was maintained. There was no significant decrease at 7 h when PP was completed.

**Figure 3 jcm-10-02969-f003:**
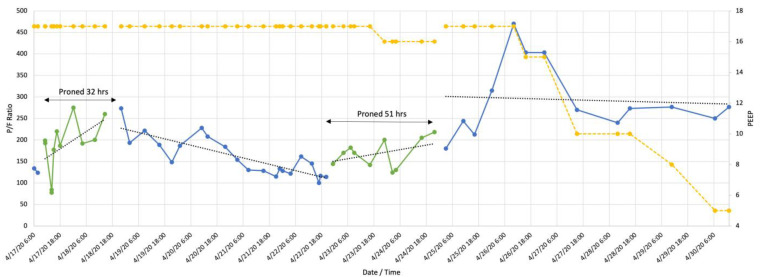
Patient with repeated prone positioning. Following an initial P/F ration of 124 (blue line) the patient was turned prone (green line) for 32 h when P/F 260 was attained. There was a steady decline to P/F 100 at which point he was proned for 51 h after which time the P/F ratio remained stable. Yellow line indicates positive end expiratory pressure (PEEP).

**Figure 4 jcm-10-02969-f004:**
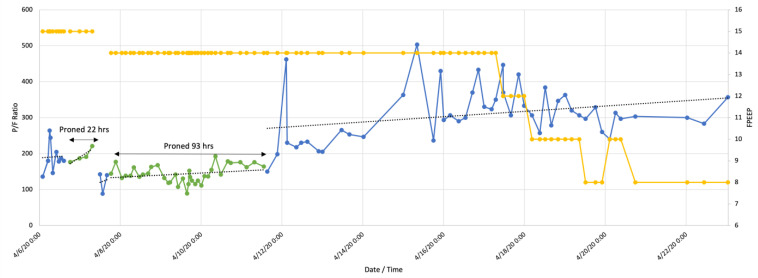
Patient with repeated prone positioning. Following a P/F ratio of 176 (blue line), the patient was turned prone for 22 h (green line), turned supine at P/F 142, and returned to the prone position for 93 h. When turned supine again, there was a consistent trend of improving oxygenation. Yellow line indicates positive end expiratory pressure (PEEP).

**Figure 5 jcm-10-02969-f005:**
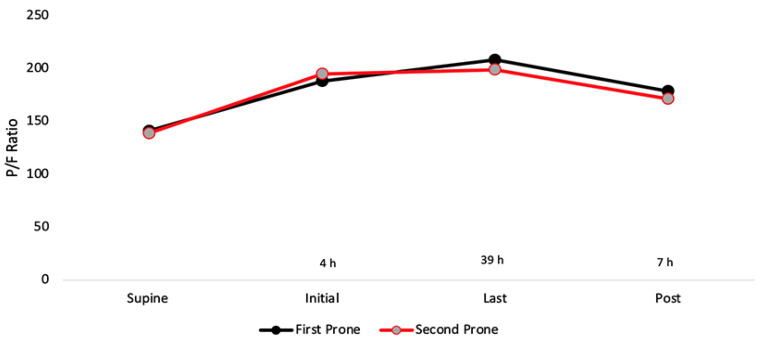
Time course change in P/F ratio for reproning. Supine represents immediately before PP. Post represents 7 h after completing PP. Reproning is not necessary to improve oxygenation compared to an initial prone.

**Figure 6 jcm-10-02969-f006:**
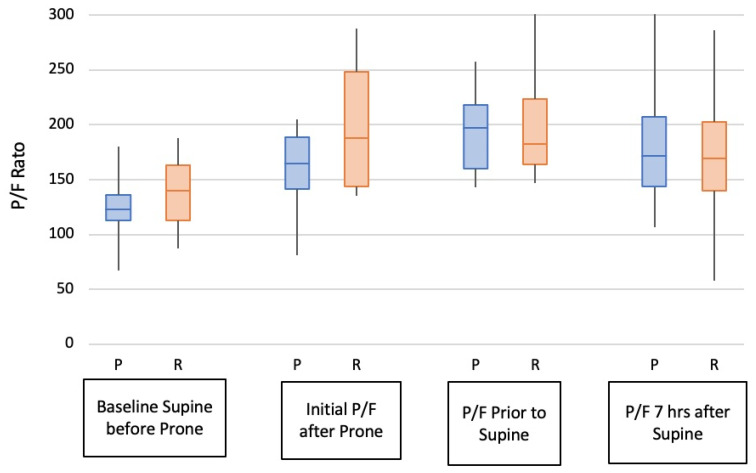
P/F ratio comparison between prolonged (P) and repeated (R) prone positioning. Prone P/F ratio prior to turning supine compared to baseline, *p* < 0.001; P/F for repeated prone compared to baseline, *p* < 0.004; P/F 7 h after return to supine for single prone, *p* < 0.019 compared to baseline; P/F 8 h after return to supine from multiple prone turns, *p* < 0.18.

**Figure 7 jcm-10-02969-f007:**
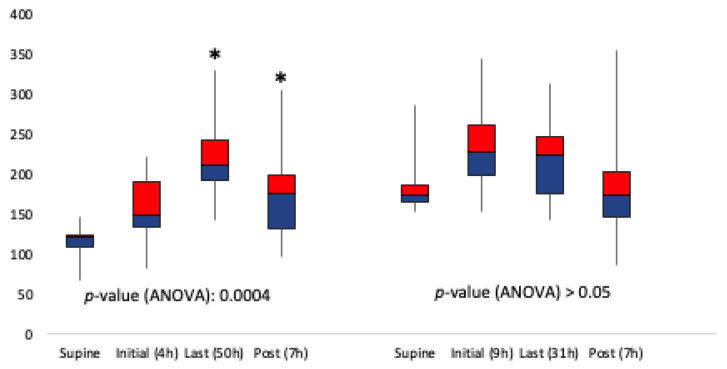
Time course change for P/F ratio of later proning versus early proning. Supine is immediately before PP. Post represents 7 h after completing PP. There was statistical significance (asterisks) in initiating prone positioning when the P/F ratio decreased to ≤150 (defined as later proning), and there was no statistical significance in initiating prone positioning when the P/F ratio was >150 (defined as early proning).

**Table 1 jcm-10-02969-t001:** Summarizes the timeline from symptom onset to the first prone positioning.

Timeline (Days): Median (IQR) of Patients Positioned Prone (PP)				
Event	Prolonged PP	Repeated PP	Early PP	Late PP
Symptom Onset to Hospital Admission	6 (4–9)	5 (3–7)	5 (4–7)	6 (4–8)
Hospital Admission to SICU Admission	0 (0–1)	0 (0–1)	0 (0–2)	1 (0–2)
SICU Admission to Intubation	0 (0–0)	0 (0–0)	0 (0–0)	0 (0–0)
Intubation to First Prone Positioning	3 (1–5)	4 (3–4)	4 (3–4)	4 (1–5)

**Table 2 jcm-10-02969-t002:** Characteristics of Proned Patients.

	Prolonged Prone	Repeated Prone	Early Prone	Late Prone
	PP (*n* = 12)	PP (*n* = 12)	PP (*n* = 10)	PP (*n* = 10)
Age, years	63.3 ± 11.5	56.6 ± 16.3	60.9 ± 19.6	57.1 ± 22.1
Men, *n* (%)	9 (75%)	10 (83%)	6 (60%)	7 (70%)
Severity of ARDS				
Mild, *n* (%)	0 (0%)	0 (0%)	0 (0%)	0 (0%)
Moderate, *n* (%)	4 (33%)	4 (33%)	6 (60%)	3 (30%)
Severe, *n* (%)	8 (67%)	8 (67%)	4 (40%)	7 (70%)
Treatment in ICU				
Vasopressor, *n* (%)	12 (100%)	12 (100%)	10 (100%)	10 (100%)
Inhaled nitric oxide, *n* (%)	7 (58%)	6 (50%)	3 (30%)	5 (50%)
Hospital Mortality, *n* (%)	4 (33%)	4 (33%)	4 (40%)	3 (30%)
Duration of PP (h): median (IQR)	57 (45–66)	117 (97–153)	84 (46–112)	96 (57–142)
ICU Length of Stay (days): median (IQR)	20 (17–33)	20 (17–39)	20 (14–25)	19 (16–33)
Mechanical Ventilation (days): median (IQR)	17 (15–18)	17 (17–21)	17 (16–20)	16 (14–17)

**Table 3 jcm-10-02969-t003:** Ventilation and oxygenation variables throughout the first 39 h of the prone positioning.

	Supine	1 h	7 h	12 h	24 h	32 h	39 h	*p* Value
FiO_2_	0.60	0.65	0.60	0.57	0.55	0.57	0.53	0.045
PEEP (cmH_2_O)	13	13	13	13	13	13	13	0.230
Pplat (cmH_2_O)	25.6	25.6	25.8	25.8	25.6	25.9	25.9	0.910
Cstat (mL/cmH_2_O)	27	28	28	28	28	29	29	0.834
PaO_2_/FiO_2_ (mmHg)	130	175	170	184	179	198	193	0.000

FiO_2_, fraction of inspired oxygen; PEEP, positive end-expiratory pressure; Pplat, plateau pressure; ΔP, driving pressure; Cstat, static compliance.

**Table 4 jcm-10-02969-t004:** Ventilation and oxygenation variables for repeated prone positioning.

First Prone						Reprone				
	Supine	Initial	Last	Post	*p* Value	Supine	Initial	Last	Post	*p* Value
FiO_2_	0.65	0.69	0.56	0.57	0.057	0.59	0.61	0.57	0.60	0.872
PEEP (cmH_2_O)	13	14	14	14	0.774	13	14	14	14	0.125
Pplat (cmH_2_O)	25.7	25.7	25.3	24.6	0.041	24.5	24.5	25.5	25.5	0.018
Cstat (mL/cmH_2_O)	31	33	35	39	0.179	38	38	35	38	0.199
PaO_2_/FiO_2_ (mmHg)	141	188	208	179	0.011	139	195	199	171	0.004

Reprone, the next prone session; FiO_2,_ fraction of inspired oxygen; PEEP, positive end-expiratory pressure; Pplat, plateau pressure; ΔP, driving pressure; Cstat, static compliance.

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
