# Peer review of "Efficiency of Prolonged Prone Positioning for Mechanically Ventilated Patients Infected with COVID-19"

_jcm, 2021, doi:10.3390/jcm10132969_

Round 1

Reviewer 1 Report

Thank you very much for giving me the opportunity to review the manuscript entitled “Efficiency of Prolonged Prone Positioning for Patients Infected 2 with COVID-19”. The authors demonstrate the efficacy of prone positioning for patients with COVID-19-induced ARDS. There are several concerns in the manuscript.

  1. There are a lot of grammatical errors throughout the manuscript, especially in the abstract. English editing must be necessary.
  2. The inclusion criteria and exclusion criteria for the study participants are not clear. How many patients with CARDS were admitted to the ICU during the study period?
  3. Were the results of arterial blood gas analysis at all the time points obtained from all the included patients?
  4. Were there any patients who received a single PP management for less than 39 hours.
  5. In the Figure.1, it is not clear which pairs of the values were directly compared in the post-hoc analysis after ANOVA.
  6. It is not clear whether the improved P/F ratio compared with the baseline observed after turning back to supine position is due to the PP management or not. The P/F improvement might be caused by the natural recovery over time. These points must be discussed.
  7. The results do not support the prolonged PP management (>39 h) is superior to PP management for less than 39 hours. Moreover, it is not clear whether the prolonged PP management is not inferior to the repeated PP managements. Direct comparison of these managements (i.e., prolonged vs. short and prolonged vs. repeated) should be performed. If it is not possible, these points must be stated as limitations.
  8. The comparison of patients with the PP management and those with the supine management will strength the authors conclusions.

Reviewer 2 Report

MAJOR COMMENTS

  1. Several reports suggest that COVID-19 associated ARDS behaves different to conventional ARDS. This seems to be a function of time related to onset of the disease, with a initial phase where respiratory system compliance is essentially normal, and hypoxemia is rather caused by endothelial lesions and perfusion disorders caused by vascular thrombosis. In a later phase, inflammation causes lung collapse, impaired compliance, and consolidation of lung tissue. Thus, theoretically there might be different physiological effects/mechanisms depending on the phase of CARDS, with potential for alveolar recruitment in the later phase. Based on these considerations, 1) the authors provide the times between symptom onset, hospitalization, ICU admission, begin of mechanical ventilation, and first prone positioning. 2) Time to PP could also be a cofactor for the analysis? Does it correlate with positive response? 3) the issues should be briefly discussed.
  2. Statistics: I suggest performing a linear model analysis with cofactors time from onset of symptoms, time of PP, FIO2 before PP…
  3. Results, line 83: The selection criteria for groups should be moved to the Methods. In addition, it is suggested here that oxygenation is a function of time in CARDS, which is not necessarily true (see above #1 and Gattinoni’s concept of l-type and h-type.)
  4. 1+2: I suggest showing lines of single patients. This would allow to visually distinguish between responders and non-responders.

MINOR COMMENTS

  1. Line 49: check text seize.
  2. Line 61: typo „involved“

Reviewer 3 Report

Comments to authors

The authors have conducted a retrospective cohort study of adults admitted to critical care with COVID-19, to investigate the effects of proning on hypoxaemia.

Major limitations

  1. Retrospective study design- the authors need to discuss the limitations associated with retrospective study designs in their ‘Discussion’ section. There is a considerable risk of introducing various types of bias in such a design, including selection bias, and the reader needs to be made aware of this. There is also the risk of confounding due to known and unknown factors that may be associated with the measured outcome (hypoxaemia).
  2. Why were the specific time points of 1, 7, 12, 24, 32 and 39 hours selected? Could the authors explain the rationale for this in the ‘Methods’ section?
  3. Could the authors adjust their outcome for other potential confounders that may alter the disease course in a patient with COVID-19, such as COVID-19 specific treatments (Dexamethasone, Tocilizumab, Remdesivir etc), adjunctive therapy (antimicrobial use etc), disease severity and demographics? The value of their findings would certainly be augmented, if their results could be adjusted for potential confounders.
  4. The authors should report the rate of any adverse effects (ET tube displacement, pressure areas etc) they encountered that were associated with proning. Was there a difference in these adverse events between the prolonged proning versus other groups?
  5. Could the authors present the baseline demographics, presence of comorbidities, disease severity and treatment details for each of the comparator groups they have analysed (prolonged proning, early proning and repeated proning groups), as this would allow meaningful interpretation of the results?
  6. Could the authors present adjusted outcomes (mortality, ICU length of stay) for patients in each of the comparator groups, if possible? This would make the study more clinically relevant, as we do not yet know the significance of this observed improvement in hypoxaemia in relation to clinical outcomes. I appreciate that statistical comparisons may prove to be difficult due to the low event numbers, but at least this data should be presented.

Minor comments

  1. Could the authors ensure that all abbreviations are initially described (for example CARDS in the introduction)?
  2. The first paragraph in the ‘Results’ section should be moved to the ‘Methods’ section

Round 2

Reviewer 2 Report

Thanks for the revision. No further comments.